# Bioactive Functions of Lipids in the Milk Fat Globule Membrane: A Comprehensive Review

**DOI:** 10.3390/foods12203755

**Published:** 2023-10-12

**Authors:** Junyu Pan, Meiqing Chen, Ning Li, Rongwei Han, Yongxin Yang, Nan Zheng, Shengguo Zhao, Yangdong Zhang

**Affiliations:** 1Key Laboratory of Quality & Safety Control for Milk and Dairy Products of Ministry of Agriculture and Rural Affairs, Institute of Animal Sciences, Chinese Academy of Agricultural Sciences, Beijing 100193, China; buckyneng@foxmail.com (J.P.); mqchen1997@163.com (M.C.); zhengnan@caas.cn (N.Z.); zhaoshengguo@caas.cn (S.Z.); 2College of Food Science and Engineering, Qingdao Agricultural University, Qingdao 266109, China; lining20211114@163.com (N.L.); qauhan@qau.edu.cn (R.H.); qauyang@qau.edu.cn (Y.Y.)

**Keywords:** milk fat globule membrane, phospholipids, ganglioside, cholesterol, bioactive function

## Abstract

The milk fat globule membrane (*MFGM*) is a complex tri-layer membrane that wraps droplets of lipids in milk. In recent years, it has attracted widespread attention due to its excellent bioactive functions and nutritional value. *MFGM* contains a diverse array of bioactive lipids, including cholesterol, phospholipids, and sphingolipids, which play pivotal roles in mediating the bioactivity of the *MFGM*. We sequentially summarize the main lipid types in the *MFGM* in this comprehensive review and outline the characterization methods used to employ them. In this comprehensive review, we sequentially describe the types of major lipids found in the *MFGM* and outline the characterization methods employed to study them. Additionally, we compare the structural disparities among glycerophospholipids, sphingolipids, and gangliosides, while introducing the formation of lipid rafts facilitated by cholesterol. The focus of this review revolves around an extensive evaluation of the current research on lipid isolates from the *MFGM*, as well as products containing *MFGM* lipids, with respect to their impact on human health. Notably, we emphasize the clinical trials encompassing a large number of participants. The summarized bioactive functions of *MFGM* lipids encompass the regulation of human growth and development, influence on intestinal health, inhibition of cholesterol absorption, enhancement of exercise capacity, and anticancer effects. By offering a comprehensive overview, the aim of this review is to provide valuable insights into the diverse biologically active functions exhibited by lipids in the *MFGM*.

## 1. Introduction

Milk fat, a highly regarded and widely consumed nutrient, not only serves as a vital energy source for the human body, but also constitutes a significant reservoir of essential fatty acids and vitamins. Approximately 98% of the fat in milk exists as milk fat globules, which are mainly composed of triacylglycerols inside and surrounded by the milk fat globule membrane (*MFGM*) [1,2]. After secretion from the endoplasmic reticulum, milk fat globules are coated with a monolayer of phospholipids to form cytoplasmic lipid droplets (CLDs), a membrane derived from the endoplasmic reticulum. Subsequently, during secretion out of the cell, droplets bind to the apical plasma membrane, the outer bilayer membrane of the *MFGM* that contains a variety of polar lipids and proteins [3,4]. *MFGM*s with a thickness of 10–20 nm have a mass of 2–6% of the milk fat globules and act as a natural emulsifier, preventing fat from coalescing in the presence of enzymes [5,6].

This three-layer membrane with biologically active functions is mainly composed of proteins, enzymes, triacylglycerol, and polar lipids (cholesterol, sphingolipids, etc.) [7,8], and the composition of the *MFGM* in different species is different (Table 1). The nutrients in the *MFGM*, especially sphingolipids, phospholipids and proteins, make it an excellent source for the development of nutraceuticals, especially infant-development supplements [9,10]. The diverse bioactive functions of the *MFGM* have led to the emergence of various *MFGM*-related industrial products. Examples include Lacprodan *MFGM* 10 and Lacprodan PL 20, which serve as supplements for phospholipids [11]. The emulsifying properties of the *MFGM* help enhance the texture and flavor in various food products. For instance, the incorporation of *MFGM*s in the bread production process serves to retain moisture in bread crumbs, thereby impeding bread aging and hardening. Adding 4% *MFGM*-enrichment products combined with homogenization during the yogurt production process can increase the interaction between the *MFGM* and protein during the production process and improve the texture of yogurt [12].

Both fresh milk and by-products derived from dairy processing serve as effective sources of *MFGM*s [13]. The process of isolating an *MFGM* from milk typically involves several steps: the separation of milk fat globules, cream washing, *MFGM* release, and *MFGM* collection [6]. During the permeation process, the addition of reverse osmosis water helps maintain a constant feed rate after filtration to eliminate whey and casein. Finally, acidification is employed to isolate the *MFGM* [14]. *MFGM* separation can also be conducted by performing ceramic dia-microfiltration with a pore size of 1.4 μm on preheated whole raw milk, which can achieve the best results of a 2.5% low-fat penetration and 97% high protein penetration. Fractions of buttermilk and butter serum are separated from the filtered material, the pH is adjusted to 4.8, and an *MFGM* is obtained by centrifugation [15]. This method has simpler steps and better industrial adaptability. Industrial processes typically utilize by-product milk (e.g., buttermilk, cheese whey, etc.) to separate *MFGM*s. For instance, the removal of casein from buttermilk can be achieved through rennet-induced coagulation, followed by filtration to eliminate whey proteins. The *MFGM* is subsequently collected through diafiltration steps [16]. It must be noted that these by-products often undergo high-temperature heat treatment, leading to protein denaturation on the *MFGM* or reactions with whey protein or sugars, which can alter the emulsifying properties of the *MFGM* [17].

Dietary supplementation with *MFGM* lipids has demonstrated numerous beneficial bioactive functions. For instance, the addition of *MFGM* to milk powder has been found to mitigate the impact of phytosterols on infant nutrition by competing with cholesterol for absorption [18]. Furthermore, the intake of milk-derived phospholipids has been demonstrated to suppress the endocrine stress response in individuals exposed to high-intensity stress, with phospholipid-supplemented subjects exhibiting a faster recovery rate [19]. Furthermore, in a mouse study, ganglioside-rich *MFGM* isolates showed the potential to reduce carrageenan-induced paw edema, indicating their anti-inflammatory properties [20].

We used Google Scholar and Web of Science search engines to summarize reviews and articles between 2003 and 2023, using phospholipids, sphingomyelin, gangliosides, and *MFGM* as keywords. We chose to focus on the clinical trials focusing on the dietary supplementation of *MFGM*-enriched or isolated lipids. The purpose of this review is to offer an insightful understanding of the nutritional value of lipids in *MFGM*s, offering valuable insights into both daily dietary considerations and production practices.
foods-12-03755-t001_Table 1Table 1Content of *MFGM* components in bovine milk [21,22,23,24].Component*MFGM*Whole MilkProtein25–60%1–4%Cholesterol2%80%PL15–30%60–70%PC (% PL in *MFGM*)27.432.7PE (% PL in *MFGM*)33.028.5PS/PI (% PL in *MFGM*)17.814.1SM (% PL in *MFGM*)18.823.0PL: phospholipids; PC: phatidylcholine; PE: phosphatidylethanolamine; PS: phosphatidylserine; PI: phosphatidylinositol; SM: sphingomyelin.

## 2. Lipids in the MFGM

### 2.1. Composition and Distribution of Lipids in the MFGM

As the predominant components of monolayers, the primary lipids present in the tri-layer membrane of milk fat globules include phosphatidylethanolamine (PE), phosphatidylcholine (PC), phosphatidylinositol (PI), phosphatidylserine (PS), sphingomyelin (SM), cholesterol, and gangliosides [25,26,27,28]. The surface layer of the bilayer primarily consists of glycolipids, cerebrosides, and gangliosides, while the inner layer of the bilayer mainly contains PE, PI, and PS. Phospholipids are the predominant components of the monolayers [29].

Structurally, SM consists of long chains of bases, such as sphingosine, which form the backbone of the molecule [30]. On the other hand, glycerophospholipids are composed of phosphoric acid, glycerol, fatty acids, hydroxyl compounds, and fatty acids [31]. The presence of glycerophospholipids with relatively high unsaturation levels contributes to the improved fluidity of *MFGM*s [32]. Gangliosides, on the other hand, are formed through glycosidic linkages between ceramides and residues of sialic acid [33].

Cholesterol is found primarily in the outer bilayer of *MFGM*s and forms rigidly ordered domains known as lipid rafts when it binds with SM [34,35] (Figure 1). In contrast, the disordered phase of the membrane mainly consists of phospholipids. Lipid rafts have the ability to bind to proteins and can induce signaling processes. The structural role of lipids in *MFGM*s can be characterized by high-throughput synchrotron radiation X-ray diffraction (SR-XRD) and differential scanning calorimetry (DSC) [36]. Previous research has shown that the cholesterol enhances the order of the milk SM bilayer membrane, with the ordered phase being achieved at a 33 mol% cholesterol content. The ratio of cholesterol/SM can influence the interfacial properties of *MFGM*s, thereby impacting the functional properties of milk fat globules and their digestion mechanisms [37]. For the above-mentioned lipids, clinical trials have proven their beneficial effects on physical function improvement, development, and anticancer roles (Table 2).

### 2.2. Differences of Fatty Acid Composition in the Composition of Milk Fat Globules and MFGMs

The fatty acid compositions of *MFGM*s and MFGs (milk fat globules) were significantly different (Table 3), which was due to the fact that the main lipids in the MFG were triglycerides, cholesterol, and retinol esters [57,58], which led to the differences of their bioactive functions. Saturated fatty acids accounted for 55.2–67.0% and unsaturated fatty acids accounted for 33.0–44.8% in the *MFGM*; saturated fatty acids accounted for 66.3–73.0% and unsaturated fatty acids accounted for 27.0–33.7% in the *MFGM* [59,60,61]. There are many studies on the relative quantification of fatty acids in the *MFGM*. However, there are currently few studies on the absolute quantification of all fatty acids in the *MFGM*. This is because the fatty acid content in the *MFGM* is too low and requires higher throughput and sensitive technology for detection.

## 3. Characterization of Lipids in the MFGM

Liquid chromatography with an evaporative light-scattering detector (ELSD) is a frequently employed method for quantifying and characterizing lipids [64]. In the study by Zou et al., this technique was employed to determine the concentrations and relative proportions of PC, PI, PS, PE, and SM in the *MFGM*s of colostrum, mature milk, and transitional milk from cows. The findings indicated that the concentration of polar lipids in the total lipids reached its peak during the transitional stage of lactation. The relative content of the *SM* did not exhibit significant changes; however, the level of phosphatidylcholine in mature milk was higher compared to the other two stages [65]. This study highlighted that the level of polar lipids in mature milk was significantly higher than that in colostrum, which was in accord with the previous research [66]. In a previous comparative analysis, the lipid profile of the *MFGM* levels in buffalo and cow milk products were examined using HPLC-ELSD [60]. The results indicated that the percentage of phosphatidylcholine in buffalo milk was higher than that in cow milk, while the percentage of *SM* was lower when compared to cow milk.

Lipidomic, in combination with mass spectrometry, offers a powerful approach for identifying various lipid species based on different phospholipid classes [67]. In the study by George et al., the relationship between growth characteristics and lipids in the *MFGM* was explored, and the concentration and intake of lipids from different *MFGM*s were compared. Through LC-MS analysis, this research identified a total of 166 *MFGM* lipid species originating from 10 fractions. The study further demonstrated that infants exhibited variations in the content and intake of different lipids that existed in *MFGM*s, and the intake of *MFGM* lipids was positively correlated with infant development [68]. Similarly, Brink et al. utilized UPLC-MS to identify 338 *MFGM* lipid species derived from 10 fractions, thereby completing the lipid characterization of different commercial *MFGM* materials. In a study by Ali et al., UPLC-ESI-Q-TOF-MS was employed to identify 100 *MFGM* lipids derived from 7 fractions [69,70]. With the development of lipid analysis technology, *MFGM* lipid differences among different species have been widely characterized (Table 4), which is helpful for conducting better research on the nutritional value of milk and dairy products [37,71,72,73,74].

## 4. Various Factors Alter the Lipid Composition of MFGMs

The size of the MFG can have a significant impact on the composition of the *MFGM*. Due to the different surface areas of MFGs, the volume ratio changes, and therefore the proportion of polar lipids also changes. Furthermore, diet and lactation change the size of the MFG, which has a significant effect on the composition of *MFGM*s [59,75,76,77]. The proportion of PI in small MFGs (3.32 ± 1.21 µm) is significantly lower than that in large MFGs (7.61 ± 0.90 µm), while the proportion of PE is significantly greater than that in large MFGs [78]. The size of the MFG at different positions in the cream fraction is classified. The upper milk fat globule is named F1, which is the largest, and the lower milk fat globule is the smallest, named F6. The results of the HPLC-ELSD analysis show that there are significant differences in the ratios of PI and PC between F1 and F6 fractions with significant differences in the MFG size, while there are no significant differences among F4, F5, and F6, and the changed composition of protein and fat produce this result [79]. In these groups of MFGs, the relative content of SM does not change significantly. For fat globule size-induced changes in the *MFGM* composition, this may be due to the difference in the curvature between MFGs of different sizes, resulting in different levels of dynamic processes at the molecular level, or it may be due to the rearrangement of the apical plasma membrane composition after secretion [80]. Changes in *MFGM* composition can affect its function during processing or digestion [73]. Studying the relationship between MFG size and *MFGM* composition contributes to further research on MFG secretion and allows for the production of products with *MFGM* compositions designed to meet specific needs.

## 5. Separation of Lipids in MFGMs

Extracts containing *MFGM*s are mainly divided into *MFGM*-enriched ingredients and phospholipid extracts. The former is widely used in nutrition, while the latter is mainly used in beauty products [81]. The Folch and Mojonnier methods are general lipid-extraction methods. The Folch method extracts lipids from milk using 20 volumes of 2:1 chloroform/methanol or 4 volumes of 1:1 chloroform/methanol, while the Mojonnier method extracts lipids using a mixture of ethanol, diethyl ether, and petroleum ether. After the milk fat is separated, solid phase extraction is usually used for fractionation [82,83]. Similarly, phospholipids can be extracted from dairy by-products using switchable solvents. Tertiary amines (CyNMe2) have been proven to extract PL from raw cream, buttermilk, concentrated buttermilk, and beta-serum, and CyNMe2 can extract most of PLs (mainly phosphatidylcholine and phosphatidylinositol) from buttermilk and beta-serum, which is higher than the extraction performance of the Folch and Mojonnier methods [84,85]. The combination of ultrafiltration and supercritical fluid extraction can obtain a fraction rich in globular membrane phospholipids of milk fat [86]. Lactose and ash are removed from whey buttermilk through a 10 kDa cutoff membrane. After spray drying, the powder is subjected to supercritical fluid extraction (CO_2_, 350 bar, 50 °C). Finally, a powder containing 21% lipid (61% PL) can be obtained. It is well known that lipid removal from milk is possible using organic solvents. The use of ethanol can effectively extract phospholipids from the by-product whey protein phospholipid concentrate. Using 70% aqueous ethanol at 70 °C can obtain a lipid concentrate with a higher PL content, while using 70% aqueous ethanol at 60 °C. Ethanol can obtain a lipid concentrate with a higher SM content [87]. The research on phospholipid extraction methods is relatively mature, most methods rely on organic solvents, and have high applicability; the extracted PL enrichment can be widely used in food emulsifiers.

## 6. Phospholipids

### 6.1. The Promoting Effect of Phospholipid Supplementation on Development

The addition of *MFGM*s to infant formula has become a widespread practice due to its role in promoting cognitive development [88]. The inclusion of sphingomyelin-enriched milk in the diet of premature infants has been investigated for its impact on cognitive development [89]. In a study involving 24 very-low-birth-weight babies, the infants were separated into two groups. The trial group accepted sphingomyelin-enriched milk, where sphingomyelin accounted for 20% of all phospholipids in the milk, while the control group received milk with only a 13% sphingomyelin content [38]. At eighteen months old, the babies in the group of sphingomyelin-enriched milk presented significantly higher scores in various developmental assessments, including the behavior rating scale of the BSID-II test, the Fagan test, visual evoked potentials (VEPs), and the sustained-attention test, compared to the control group. These findings suggest that infants receiving sphingomyelin-enriched milk exhibit improved neurobehavioral development.

Previous research examining the neurocognitive development and longitudinal trajectories of the brain in children who were fed different formulas for at least 3 months found significant developmental differences among the groups. Specifically, factors, such as long-chain sphingomyelin, iron, fatty acids, folic acid, and choline, were closely associated with early myelination trajectories [40]. Another study compared the impact of formula and conventional milk products on the central nervous system of 182 preschool children. The formula milk used in the trial group contained a 8–9-times-higher phospholipid concentration compared to the non-formula milk. The trial group received 200 mL of formula milk without phospholipids, while the control group received formula milk containing 500 mg of phospholipids. The children were evaluated using the Achenbach system of empirically based assessment. The results of the research indicate that formula milk with a higher phospholipid concentration has a more pronounced effect on children’s behavioral regulation [41]. These investigations emphasize the significance of specific nutrients, particularly phospholipids, in formula milk for promoting healthy brain and behavioral developments in children. The inclusion of appropriate levels of these nutrients in formula milk can positively impact neurodevelopment. This positive effect can be attributed to the role of sphingomyelin as an important component of the myelin sheath, as indicated by the previous research [39]. As a merely structural component of myelin, sphingomyelin promotes the proliferation, maturation, and differentiation of oligodendrocyte precursor cells (OPCs), as well as increased axonal myelination [90,91]. This explains why the dietary supplementation of *MFGM*-derived sphingolipids can increase cognitive and developmental functions in subjects. Considering the complexity of the nutritional needs of infants during their development, simple formula milk powder can no longer meet the market’s demands, and phospholipid supplementation is an effective means to enhance product competitiveness.

### 6.2. The Promoting Effect of Phospholipid Supplementation on Memory

Indeed, the supplementation of PS and sphingomyelin has been shown to enhance memory function [19]. In previous research involving 75 males aged 30 to 51 years old, different test groups were given milk containing a placebo, 0.5%, and 1% phospholipids (phospholipid content adjusted using Lacprodan PL 20) over a period of 42 days. The stress-protective effects of the phospholipids were evaluated using the Trier social stress test. It was concluded that high doses of phospholipids could inhibit the activity and responsiveness of the hypothalamic–pituitary–adrenal axis (HPAA) and result in a blunted psychological stress response. The age of the subjects and the duration of phospholipid supplementation may influence these effects [42]. In a study involving piglets, Lacprodan PL-20 (0%, 0.8%, 2.5% *v*/*v*) was added to the diets of three groups of piglets. The development of the piglets was observed from days 2 to 28 postpartum. The piglets’ performance in the T-maze was measured on day 14, and brain MRI data were obtained on day 28 postpartum. The piglets supplemented with 0.8% and 2.5% gangliosides showed better performances in the maze, had higher brain weights, and exhibited more white and gray matter. This suggests that gangliosides enhance spatial learning in newborn piglets and influence brain development [43]. Similar findings were observed in mouse experiments, where the supplementation of phospholipids from the *MFGM* improved the memory of mice [92]. These investigations emphasize the potential cognitive benefits of phospholipid supplementation, including an improved capacity to memorize and spatial learning, in both human and animal testing methods. The specific effects may vary depending on factors, such as dosage, duration of supplementation, and the age or developmental stage of the subjects. A recent study demonstrated the promoting effect of sphingomyelin on hippocampal development. Sphingomyelin can enter the nucleus of hippocampal cells, causing it to overexpress the sphingomyelin phosphodiesterase 4 gene encoding a neutral sphingomyelinase, thereby promoting changes in the soma of hippocampal cells and the formation of synapses [93]. In addition, the dietary supplementation of *MFGM* can also change the lipid abundance in the hippocampus without changing the lipidome of other brain tissues [94], which may also be the reason why supplementation with *MFGM*s improves memory, and the specific reason needs further research.

### 6.3. The Promoting Effect of Phospholipid Supplementation on Exercise Performance

The supplementation of dietary phospholipids, such as PC and PS, has been shown to improve the exercise performance of humans [44,95]. The supplementation of phospholipids and sphingolipids, in combination with exercise, has been investigated for its potential to improve muscle movement and neuromuscular development, including the formation of neuromuscular junctions. In Yoshinaka’s study, 71 subjects were divided into two groups. One group ingested 1 g of *MFGM* placebo (167 mg/placebo), while the other group received a placebo in the form of whole milk powder. Tablets containing *MFGM*s were produced for the test group, and placebo tablets were composed of whole milk powder. Both groups engaged in low-intensity exercise, and the trial lasted for eight weeks. The test group demonstrated a better performance in foot tapping and opening and closing steps, which could be attributed to the presence of sphingomyelin, one of the components that promote the development of nerve and muscle fibers [45]. Kim’s research expanded on this by including a placebo plus exercise group and an *MFGM* plus exercise group. This design allowed for a more targeted study to determine whether supplementation with milk fat globules or daily low-intensity exercise alone could improve frailty in the subjects [46]. These works suggest that dietary supplementation with phospholipid and sphingolipid supplements, in conjunction with exercise, may have positive effects on neuromuscular development and muscle movement. The combination of the phospholipid supplementation of *MFGM* and exercise can promote the expression of docking protein-7 and myogenin mRNA, thereby promoting the formation of neuromuscular junction synapses [11]. In addition, previous research points out that the improvement in exercise performance caused by phospholipid supplementation is due to the fact that phospholipids stabilize the cell membrane of red blood cells, thereby improving the oxygen transport capacity of red blood cells [96].

### 6.4. MFGM Phospholipid Supplementation Helps Alleviate Alzheimer’s Disease

Alzheimer’s disease is a neurodegenerative disease caused by cognitive decline with age, usually in patients over 65 years old [97]. As an important component of the cell membrane, changes in phospholipids at the cellular level cause different pathogenic processes, which can be improved by the dietary supplementation of PL [98,99]. Phospholipid supplementation by the intake of *MFGM*-enriched substances can effectively improve age-induced cognitive decline. A study on aged Wistar rats showed that dietary supplementation with enriched *MFGM*s isolated from buttermilk enhanced the rat’s spatial working memory. This was caused by changes in the lipid composition of the synaptic membrane. The contents of PS, PE, and SM increased after supplementing phospholipids, and these lipids were involved in the decline in cognitive function [47]. Recent studies indicate that AD symptoms can be effectively alleviated by adding phospholipid-rich protein powder (PP) to the diet of triple-transgenic AD (3 × Tg-AD) mice. This is because protein powder avoids neuroinflammation through the peroxisome proliferator-activated receptor γ (PPAR γ)–nuclear factor-κB signaling pathway [48].

### 6.5. Phospholipids Have a Regulating Effect on Gut Health

The dietary supplementation of phospholipids has an immunomodulatory role in immune regulation, particularly in the regulation of gut microbial composition and inflammation [100]. Compared to drugs, dietary modifications are often more cost-effective and sustainable in preventing infant diseases [101,102]. An investigation involving 119 infants aged ≤14 days old divided them into three groups: those receiving standard infant formula, *MFGM* lipid-enriched formula (*MFGM*-L), and *MFGM* protein-enriched formula (*MFGM*-P). The research aimed to assess the prevalence of adverse reactions in infants. The data showed that infants fed with the *MFGM*-L formula had the lowest frequency of diarrhea among the three groups [49]. This finding is likely attributable to the modification of gut microbiome by phospholipids or to phospholipid-induced changes in the immune system [103]. The beneficial effects of phospholipids on the gut were further reflected in their interaction with microorganisms, such as *Lactobacillus*. The interaction between phospholipids in the *MFGM* and *Lactobacillus* has been shown to significantly influence *Lactobacillus* adhesion and enhance gut microbiota health [104]. *MFGM* phospholipids can reduce the adhesion of Lactobacillus to Caco-2/goblet cell co-cultures. The electronegativity of the bacterial cell surface can be increased by adsorption or incorporation, which further causes changes in adhesion, leading to an increase in the adhesion process. *MFGM* phospholipids can promote the function of probiotics, which can provide them with beneficial prospects in the dairy industry. Moreover, phospholipids in the *MFGM* show the ability to inhibit the growth of *Helicobacter pylori* and reduce the levels of *E. coli* and *Salmonella* enteritidis [105].

### 6.6. Phospholipids Regulate Cholesterol Metabolism

Increased serum cholesterol is a major contributor to cardiovascular disease (CVD), and supplementation with milk-derived lipids has been shown to modulate cholesterol levels [106]. Lipids present in the *MFGM*, particularly sphingomyelin, have been found to inhibit cholesterol absorption in the gut [107]. Dietary sphingomyelin has a substantial impact on plasma and tissue cholesterol levels. As shown in Figure 2, milk-derived sphingomyelin exhibits a stronger effect in inhibiting rat cholesterol absorption, and this can be attributed to the compatibility of sphingosine and N-acyl groups in the presence of cholesterol, promoting their mutual attraction [50]. In a study involving 34 subjects with low-density lipoprotein cholesterol (LDL-C) levels below 5.0 mmol/L, the effects of dietary supplementation with chocolate-flavor buttermilk or placebo were investigated. The placebo formulation matched the macro/micronutrient content of the buttermilk, except for the nutrients from *MFGM*s. The research found that dietary supplementation with buttermilk containing a high concentration of milk globular phospholipids effectively inhibited cholesterol absorption [51]. The strong affinity of milk-derived sphingomyelin for cholesterol can effectively lower the cholesterol thermodynamic activity and reduce the monomer content between cholesterol micelles, thereby inhibiting cholesterol absorption [108]. These works indicate that supplementation with milk-derived lipids, particularly sphingomyelin, can have a beneficial impact on cholesterol absorption and may help in managing serum cholesterol levels.

In a test performed by Rosqvist et al., a single-blind randomized controlled experiment was conducted over eight weeks on 57 overweight participants. The test group was supplemented with 40 g of whipping cream, which served as a source of *MFGM*s because of its enriched phospholipid level and relatively complete *MFGM* structure. The dietary phospholipid concentration in the experimental group was approximately 19-fold higher than that in the control group. The results indicated that blood lipids and the LDL-C of participants without the supplementation of *MFGM*s were significantly higher. The ingestion of *MFGM*s did not increase the cholesterol concentrations, potentially due to the influence of milk phospholipids on lipid metabolism and hepatic gene expression, affecting intra- and inter-organ lipid distributions [52]. Phospholipids have the ability to interfere with specific interactions in the gut, leading to the inhibition of cholesterol absorption without interfering with the gut microbiota [53]. This suggests that the mechanism of the phospholipid regulation of cholesterol absorption involves specific interactions in the gastrointestinal tract, which effectively reduce the uptake of cholesterol without disrupting the balance of gut microbial communities. The reason why phospholipid intake inhibits cholesterol absorption may be that phospholipids inhibit the expression of transporters related to cholesterol absorption, or reduce the solubility of cholesterol and increase the size of micelles and of bile acid binding capacity [109]. Many studies have proved the inhibitory effect of phospholipids on cholesterol absorption; however, further research is needed on the specific mechanism.

### 6.7. Anticancer Effects of Dietary Phospholipids

To assess the anticancer potential of sphingolipids, the AIN-76 feed was supplemented with anhydrous milk fat (AMF) and a combination of AMF and *MFGM* (1:1). A microscopic analysis revealed a significantly reduced number of colonic lesions in mice fed with *MFGM*s compared to those fed AIN-76 or anhydrous fat. This work proved that the anticancer function was attributed to sphingomyelin in *MFGM*s [54], which was supported by the subsequent research [110]. Buttermilk, obtained by extracting lipid components using food-grade solvents, has shown promising inhibitory effects on cancer cell activity. The presence of phospholipids in the extracts appeared to play a crucial role [111]. In one study, lipids from buttermilk powder were extracted by both food-grade ethanol and a non-food-grade solvent, such as a dichloromethane–methanol solution. The extracts were then fractionated using flash chromatography and the resulting solution’s inhibitory effects on nine human cancer cell lines were evaluated using an absorbance microplate reader [112]. The aforementioned findings suggest that sphingolipids, particularly those derived from *MFGM*s and buttermilk, can possess anticancer properties and exhibit inhibitory effects on colon cancer development and progression. Further research is needed on the mechanism by which phospholipids inhibit cancer. One review concludes that this may be because phospholipids are involved in the Kennedy pathway, and perturbations regulated by this pathway are related to a variety of diseases, including cancer [113].

## 7. Gangliosides

### 7.1. Gangliosides Promote Brain Development

The supplementation of gangliosides, a type of complex glycosphingolipids, has been found to promote neurodevelopment and cognitive function, potentially due to the presence of sialic acids in gangliosides, which play a role in synaptic growth and memory formation. When infants were fed formula with an increased ganglioside content, it was noted to have a favorable influence on their cognitive development [55]. In a study involving 60 infants aged 2–8 weeks old, one group was fed standard formula milk powder while the other group was fed formula milk powder enriched with compound milk fat to adjust the ganglioside content to 9 mg/100 g [56]. The cognitive development of the infants was assessed and the group receiving the ganglioside-enriched formula showed improved outcomes. Moreover, the prenatal supplementation of gangliosides by pregnant women was also found to promote brain development in their offspring [114]. This effect may be attributed to ganglioside supplementation enhancing brain region-specific increases in astrocytes, thereby increasing plasticity in the hippocampus, which is involved in learning and memory functions [115]. The abovementioned research underlines the benefits of ganglioside supplementation in promoting neurodevelopment, cognitive function, and brain plasticity in infants and offspring. At present, as an important part of the neuronal cell membrane, the supplementation of gangliosides promotes development to a certain extent; however, the physical fitness of the subjects needs to be considered.

### 7.2. Inhibitory Effect of Gangliosides on Intestinal Pathogenic Microorganisms

Indeed, dietary sphingolipids have been shown to modulate intestinal inflammation by influencing the gut microbiota [116]. Lee et al. studied the catabolism of gangliosides from milk-derived gangliosides using nano-HPLC Chip Q-TOF MS. *Bifidobacterium infantis* and *B. bifidum* significantly decreased the levels of degraded gangliosides GM3 and GD3, while *B. longum subsp. longum* and *B. animalis subsp. lactis* did not [117]. The sialic acid produced by the degradation of gangliosides leads to changes in glycolipid distribution in the intestine and exerts a prebiotic effect on the intestine, which may explain the improvement of intestinal health by gangliosides. In vitro experiments on Caco-2 cells show that gangliosides (GM3, GD3, and GM1) and sialic acid (Neu5Ac) can effectively prevent the adhesion of diarrheal pathogens [118]. They compete with pathogens to adhere to cells and can detach pathogens that adhere to cells. GM3 and GD3 are located on the apical and basolateral membranes of the Caco-2 cells, which may facilitate further studies on the competitive adhesion of gangliosides. *MFGM* gangliosides also protect tight-junction proteins [119]. By supplementing gangliosides, the level of the anti-inflammatory cytokine interleukin-10 can be increased, thereby preventing the decline in the level of intestinal tight-junction proteins and ensuring the good health of the intestinal tract. Similarly, gangliosides control the occurrence of necrotizing enterocolitis by regulating the levels of vasoactive mediators and pro-inflammatory factors [120].

## 8. Conclusions

This article presented an overview of the types and characterization methods of the main lipids from *MFGM*s. It also summarized the current works on the biological activities and functions of these lipids, including study designs, experimental results, and functional mechanisms. The supplementation of lipids derived from *MFGM*s in the diet was shown to promote development and improve immunity, making it a potential bridge between formula and breast milk products. The article emphasized the significance of studying *MFGM* lipids in the context of food nutrition and clinical applications. While there is existing research on the topic, the article suggests that there is still much to be explored, indicating potential future research directions. It was noted that the composition of the *MFGM* was influenced by external factors, such as processing methods and the stage of lactation. Therefore, accurately separating and purifying the *MFGM* and lipids in it is crucial for their commercial application. Overall, the article emphasizes the function of lipids in the *MFGM* in various aspects of human health and nutrition. It calls for further research to better understand their functions, explore their potential applications, and develop effective methods for their utilization in the food industry.

## Figures and Tables

**Figure 1 foods-12-03755-f001:**
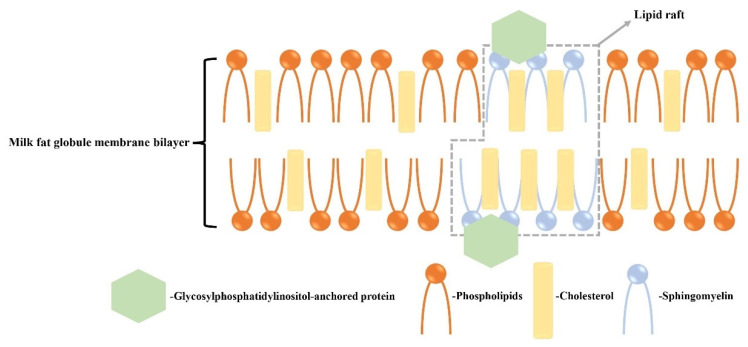
Structure of lipid rafts of *MFGM*s on cholesterol. Cholesterol in the *MFGM* binds to sphingolipids in the outer bilayer to form rigidly ordered domains, i.e., lipid rafts, and the disordered phase consists of phospholipids. The lipid rafts bind to proteins and can act as an induced signal [34,35].

**Figure 2 foods-12-03755-f002:**
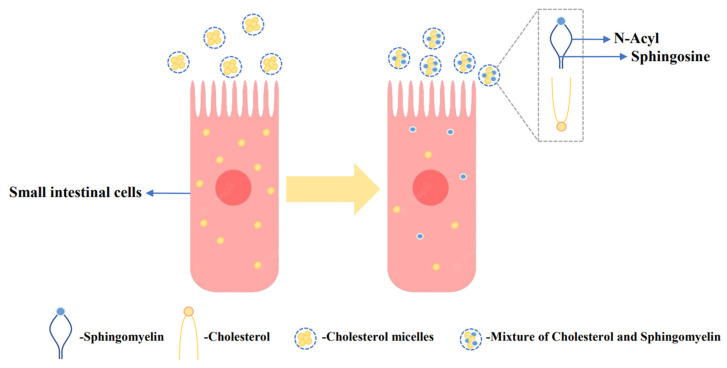
Inhibitory effect of sphingomyelin of the *MFGM* on cholesterol. In the presence of cholesterol, the mutual matching of sphingomyelin and N-acyl groups promotes the mutual attraction of cholesterol and sphingomyelin, thereby inhibiting cholesterol absorption [50].

**Table 2 foods-12-03755-t002:** Bioactive functions of lipids in the milk fat globule membrane.

Lipid	Participants	Dose	Time	Results	Reference
Sphingomyelin	Low-birth-weight preterm babies	20% of total phospholipids in milk	18 months	Supplementation of sphingomyelin in milk has a positive effect on the neurobehavioral development of low-birth-weight preterm infants.	[38]
Sphingomyelin	Wistar rats	810 mg/100 g Sphingomyelin/diet	28 days	Sphingomyelin contributes to myelination in developing rats.	[39]
Sphingomyelin and phosphatidylcholine	Children aged 0–5 years	62 mg/L 85 mg/L	90 days	Sphingomyelin and phosphatidylcholine have significant effects on neural and cognitive developments.	[40]
Phospholipids	Healthy preschool children aged 2.5 to 6 years	250 mg/100 mL	6 months	High phospholipid concentration in milk is beneficial to children’s behavior regulation and the frequency of fever is significantly reduced.	[41]
Phospholipids	75 chronically stressed men aged 30 to 51 years	250 mL fat-reduced cream powder derived from bovine milk with 0.5%, 1% PL/day	42 days	Supplementation of PL increases the availability of cortisol in subjects and attenuates memory decline.	[42]
Phospholipids and gangliosides	Piglets	0.8 or 2.5% Lacprodan PL-20	26 days	Supplementation with gangliosides and phospholipids improves spatial learning in piglets and affects brain development.	[43]
Phospholipids and exercise	Seniors aged 71–75 years	1 g tablet containing 16% of phospholipid *MFGM*s per day	8 weeks	Participants taking globular membrane tablets perform better in tapping and stepping.	[44]
Phospholipids, sphingolipids, and exercise	15-week-old male SAMP1 and ICR rats	356 ± 9 mg/day diet (contain 16.6% phospholipids)	28 weeks	*MFGM* combined with exercise can improve muscle function deficits.	[45]
Phospholipids and exercise	Older women aged 82–84 years	1 g milk fat globule membrane tablet per day	12 weeks	Exercise and phospholipid supplementation may improve frailty in older adults.	[46]
Phospholipids	30 fifteen-week-old Wistar rats	0.5 g buttermilk cookie/day	4 months	Buttermilk supplementation alters synaptic membrane lipid composition and delays cognitive decline with age.	[47]
Phospholipids	twenty 3 × Tg-AD mice and 10 wild-type mice	3.4 g whey protein powder/kg/day	3 months	Supplementation of phospholipid-rich protein powder in the diet can alleviate AD symptoms.	[48]
Phospholipids	Infants aged ≤14 days	647 mg/L	4 months	Diarrhea, vomiting, ear infections, conjunctivitis, and eczema are significantly reduced in infants fed the milk fat globule membrane phospholipid formula.	[49]
Sphingomyelin	Male Sprague Dawley rats	19.5 ± 1.4% dose	7 weeks	Compared with egg-origin sphingomyelin, milk-origin sphingomyelin has a stronger effect on inhibiting the absorption of fat and cholesterol in the rat intestinal tract.	[50]
Phospholipids	Men and women with serum low-density lipoprotein cholesterol (LDL-C) <5.0 mmol/L	187.5 mg/day	8 weeks	The intake of phospholipids reduces cholesterol levels in the body, mainly by inhibiting the absorption of cholesterol in the gut.	[51]
Phospholipids	Overweight men and women	40 g/day	8 weeks	Milk-derived phospholipids significantly reduce fasting and postprandial plasma cholesterol concentrations. Milk fat enclosed by MFGM does not impair lipoprotein profiles.	[52]
Phospholipids	Menopausal women	0.3, 0.5 g/day	4 weeks	Phospholipids may reduce specific interactions involved in cholesterol absorption in the gut.	[53]
Sphingomyelin	Male Fischer-344 rats	0.11% *w*/*w*	13 weeks	Diets containing sphingomyelin are protective against colon cancer in Fischer-344 rats.	[54]
Sphingomyelin and phosphatidylserine	Healthy men with an average age of 41.5 years	13.5 g/day	3 weeks	High doses of phospholipids can dampen the activity and reactivity of the hypothalamic–pituitary–adrenal axis (HPAA) and produce in the subject a blunted psychological stress response.	[19]
Ganglioside	Wistar rats	0.2%, 1.0% CML	80 days	Dietary gangliosides benefit cognitive development in infants.	[55]
Ganglioside	Infants aged 2 to 8 weeks	11~12 μg/mL	16 weeks	Formula with increased ganglioside content in the diet is beneficial for cognitive development in healthy infants aged 0–6 months.	[56]

**Table 3 foods-12-03755-t003:** Differences in relative content fatty acid composition between the MFG and *MFGM* [27,59,60,61,62,63].

Fatty Acid	MFG	*MFGM*
Saturated fatty acids	66.3–73.0%	55.2–67.0%
Unsaturated fatty acids	27.0–33.7%	33.0–44.8%
Omega-6 unsaturated fatty acids		
C18:2 c9, t11 (CLA)	0.42–0.92%	6.81–7.37%
C18:2 c9, c12 (n-6)	1.37–1.59%	4.13–5.11%
C20:3 c8, c11, c14 (n-6)	0.07–0.08%	0.38–0.57%
C20:4 c5, c8, c11, c14 (n-6)	0.09–0.10%	<0.04%
Omega-3 unsaturated fatty acids		
C18:3 c9, c12, c15 (n-3)	0.26–0.61%	0.43–1.65%
C20:3 c11, c14, c17 (n-3)	0.11–0.12%	0.47–0.56%
C20:5 c5, c8, c11, c14, c17 (n-3; EPA)	0.03–0.04%	0.13–0.86%
C22:5 c4, c7, c10, c13, c16, c19 (n-3; DPA)	0.06–0.10%	0.32–0.56%
C22:6 c4, c7, c10, c13, c16, c19 (n-3; DHA)	0.00%	0.01–0.48%

**Table 4 foods-12-03755-t004:** Relative proportion of lipids in *MFGM*s in different species [37,71,72,73,74].

Polar Lipids (%)	Bovine	Goat	Human	Sheep	Yak
PI + PS	16.29–18.96	3.00–23.40	20.81–22.21	6.60–16.9	15.56
PC	25.74–33.12	27.00–32.00	24.39–25.08	24.50–30.50	23.18
PE	23.42–33.76	20.00–42.00	12.48–25.33	30.50–43.00	28.20
SM	24.87–25.40	16.00–30.00	29.28–40.18	22.30–28.20	33.06

## Data Availability

Data is contained within the article.

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
