# Peer review of "Bioactive Functions of Lipids in the Milk Fat Globule Membrane: A Comprehensive Review"

_foods, 2023, doi:10.3390/foods12203755_

Round 1

Reviewer 1 Report

Τhe aim of this review is to provide valuableinsights into the diverse biologically active functions performed by lipids in milk fat globule membrane.Bioactive Functions of Lipids in MFGM:

The review is limited and refers more to the results of clinical studies, many of which have not been thoroughly investigated for their effectiveness and reliability.

There is also no mention of the fatty acid composition of the membrane lipids, which is different from the lipids from the core of the fat globules, such as saturated unsaturated fatty acids, ω3, ω6 lipid indicators, etc.

As the MFGM components differ in composition and structure between different MFG size groups, and these compositional differences may modulate the functionality of the MFGM, I consider that it useful to mention data on the effect of the size of the liposomes on the quantitative and qualitative composition of MFGM and, consequently, on the bioactive functions of its lipids.

A Fractionation scheme for isolating fat globule membrane fractions and their lipids should be added.

 I believe that bibliographic references should be added to both tables and figures.

Such as:

Line 78.  Table 1. Bioactive functions of lipids in milk fat globule membrane...:

line 10,  Table 2. Phospholipids in Human and Bovine Milk

Line 109: figure 1. Structure of lipid rafts Figure 2. Inhibitory effect of sphingomyelin of milk fat globule membrane on cholesterol………

line 262:  Figure 2: inhibitory effect of sphingomyelin of milk fat globule membrane on cholesterol. ……….

Reviewer 2 Report

The manuscript “Bioactive Functions of Lipids in Milk Fat Globule Membrane: A Comprehensive Review” is generally very well written and contains data of some relevance for a general readers as well as of high relevance for specialists in the topic. Although the subject of the paper could be of interest for the readers of the journal, the paper needs some corrections

Strengths of the paper:

The paper is very interesting. It addresses important issues related to the lipids that are found in the membrane of milk fat globules. The characteristics of these lipids and their biological activity were presented (among others, important issues related to the impact of these lipids on intestinal health were raised)

Weaknesses of the paper:

There is little information related to the manufacturing practice (related to the lipids in MFGM).

How were the articles that were included in the paper selected? Was the selection based on keywords only? Was there any method of selecting articles? Were there restrictions on the time range?

- Table 1 should be corrected, because the end is not visible.

- Table 2 also shows the results for goat milk. Unfortunately, I couldn't find a reference to these results anywhere in the text of the paper. Does the article apply only to bovine and human milk, or have other types of milk been considered?

 - Figure 1 and 2 - these are the same Figures. I also suggest that the paper includes a Figures related to the milk fat globule membrane bilayer (and not in additional materials).

- Lines 66 and 69 - an unnecessary gap between a word and a dot.

- Line: 165: big gap between words.

I would suggest proofreading by a professional translator.

Reviewer 3 Report

Manuscript Title: Bioactive Functions of Lipids in Milk Fat Globule Membrane: A Comprehensive Review

The topic of the review article is interesting. I believe it will be useful to science and practise. The study is well-designed, the text clear and easy to read. 

I have some comments to the authors as follows:

1. Keywords:  L 31; add "bioactive function" and delete "development" (See attached pdf)

2.  In the first place, write the full microorganism names 

3. Write the milk fat globule membrane (MFGM) abbreviated after the first place (See attached pdf)

4. Table and sub subject titles should be corrected (See attached pdf)

5. References section should be check according  to "Foods"  rules

6.  The additions and corrections I made on the pdf should be done.

Reviewer 4 Report

The authors evaluated the Bioactive Functions of Lipids in Milk Fat Globule Membrane: 2 A Comprehensive Review,

Please check all the spaces you leave between the words and do not paste the words with the references.

The review is well organized and some minor rearks ae follow.

Place Table 1 in another section and enrich the introduction.

Table 1. Please put it in horizontal page and put all the references for every line.

All the references in the text must put only in numerical order. Please check all the text about.

Please write all the references at the end according journal's guidelines.

Please remove all very old references and add new ones so that the total number is at least 120 number.

Round 2

Reviewer 2 Report

Dear Authors,

Thank you very much for correcting the paper according to my suggestions and answering the questions.

- Table 1 - no explanation of abbreviations (PC, PE, PL and information about the species these results refer to). There are explanations of these abbreviations in the text, but in my opinion .they should be explained at the beginning or in the description of the table

The abbreviation "MFG" was not explained.

Minor editing of English language required.
